# COVID-19 Vaccine Booster: To Boost or Not to Boost

**Rahul Shekhar** [1,*] , **Ishan Garg** [2] , **Suman Pal** [1] , **Saket Kottewar** [3] **and Abu Baker Sheikh** [1]

1   Health Sciences Center, Department of Internal Medicine, University of New Mexico, Albuquerque, NM 87131, USA; spal@salud.unm.edu (S.P.); absheikh@salud.unm.edu (A.B.S.)
2   Department of Internal Medicine, Maimonides Medical Center, Brooklyn, NY 11219, USA; ishangargmd@gmail.com
3   Department of Medicine, Division of Hospital Medicine, University of Texas Health San Antonio, San Antonio, TX 78229, USA; kottewar@uthscsa.edu
*   Correspondence: rshekhar@salud.unm.edu

**Abstract:** Developing safe and effective vaccines against severe acute respiratory syndrome coronavirus 2 (SARS-CoV-2) at a breakneck speed has been an exceptional human achievement. It remains our best hope of containing the coronavirus disease 2019 (COVID-19) pandemic. However, newer, more aggressive SARS-CoV-2 viral strains, as well as the possibility of fading immunity following vaccination, have prompted health officials to investigate the necessity for additional immunization. This has put further pressure on disregarded human life in lower-income countries that already have minimal access to COVID-19 vaccines. The Centers for Disease Control and Prevention (CDC) have recommended a third COVID-19 vaccine dose in immunocompromised individuals in a recent announcement. Governments and health care officials need to develop usage guidelines for COVID-19 vaccine booster doses while considering the dangers of potential waning immunity and new viral strains and prioritizing vulnerable populations everywhere, including those living in lower-income countries.

**Keywords:** coronavirus 2019; COVID-19; vaccine; COVID-19 vaccines; booster; third dose; vaccine inequity; immunocompromised

## 1. Introduction

Vaccination against severe acute respiratory syndrome coronavirus 2 (SARS-CoV-2) remains our main hope of controlling the coronavirus disease 2019 (COVID-19) pandemic. Globally, as of August 2021, there have been over 205 million confirmed cases of COVID-19, including over 4.3 million deaths [1]. An urgent need for safe and effective COVID-19 vaccines was met by a collaborative effort between the scientific community, the federal government, and pharmaceutical companies.

COVID-19 vaccines that are approved by World Health Organization (WHO) under the emergency use listing (EUL) include the messenger RNA (mRNA) BNT162b2 Pfizer BioNTech (Pfizer, Inc; Philadelphia, PA, USA) and mRNA-1273 Moderna vaccines (MOdernaTX, Inc; Cambridge, MA, USA); viral vector vaccines (AstraZeneca, Cambridge, UK) and Janssen Ad26.COV2.S (Janssen Biotech, Inc; A Janssen Pharmaceutical company, Johnson & Johnson; New Brunswick, NJ, USA); and inactivated virus vaccines Sinopharm (China National Pharmaceutical Group, Beijing, China) and Sinovac (Sinovac Biotech Ltd.; Beijing, China) [1–4]. As of 23 August 2021, the FDA granted full approval to the Pfizer-BioNTech COVID-19 vaccine to prevent COVID-19 disease in individuals who are 16 years of age and older [5].

As of October 2021, 47.9% of the world population has received at least one dose of a COVID-19 vaccine. In the United States, 57.1% of the population is fully vaccinated. However, the situation is much worse in developing countries; for instance, in low-income countries, only 2.8% of people have received at least one dose of a COVID-19 vaccine [6].

This inequitable administration of COVID-19 vaccines is an unacceptable situation from both ethical and healthcare standpoints.

Despite this global inequity in COVID-vaccination, a few countries that have vaccinated a more significant proportion of their population face a new set of questions, including the emergence of viral variants and concerns about waning immunity after vaccination.

Viruses constantly change through mutation. When a virus has one or more new mutations, it is called a variant of the original virus. Currently, several variants of the virus (SARS-CoV-2) that causes coronavirus disease 2019 (COVID-19) are causing global concerns, as some of these variants, including the B.1.1.7 (Alpha), B.1.351 (Beta), P.1 (Gamma, formerly named B.1.1.28.1), and B.1.617.2 (Delta) variants, are more transmissible and can cause more severe disease than wild-type SARS-CoV-2 virus [7].

Some early studies have also suggested that the antibody level triggered by COVID-19 vaccines is falling [8–11]. Based on available data, the CDC recommends COVID-19 vaccine booster shots for Pfizer-BioNTech and Modena vaccine recipients who have completed their initial series a minimum of six months ago and who are 65 years and older or who are adults who live in long-term care settings, who have underlying medical conditions, or who work/live in high-risk settings [12]. This article reviews the current literature on the need for a vaccine booster and its potential target population.

## 2. How Does the COVID Vaccine Work

Most COVID vaccines require two doses that are administered 3 to 12 weeks apart to provide adequate immunity to the individual, an exception being the Janssen vaccine, which is currently used as a single dose. The mRNA vaccines encode for the SARS-CoV-2 spike protein, which, when injected, is taken up and transcribed in the host cell, producing the spike protein, which is subsequently presented on the cell surface to B and T cells, resulting in an immune response [13–15]. Viral vector vaccines use safe adenovirus-based vectors that cannot cause disease but can serve as vectors to deliver genetic material from the COVID-19 virus to host cells. The host cells make copies of the coronavirus protein (spike protein) to generate an immune response, producing T-lymphocytes and antibodies against the viral antigen (spike protein) [13]. Inactivated or weakened virus vaccines use a form of the virus that has been inactivated or weakened, so it does not cause disease but still generates an immune response [13].

After the first dose, this immune response (T-lymphocytes and antibody level) slowly drops, leaving behind a small pool of memory B and T cells to guard against future attacks from the same pathogen [16]. A second dose causes a second bigger immune response, which slowly decreases over time. However, a larger number of memory B cells pool (with higher affinity for antigen-affinity maturation) are left behind, helping stage a more extensive and faster immune response against the same pathogen in the future [16–20]. The Centers for Disease Control and Prevention (CDC) recommend a second dose of the Pfizer and Moderna COVID-19 vaccines [21].

The second dose of these COVID-19 vaccines is administered three (for Pfizer) to four weeks (Moderna) after the first dose. Some studies have explored the effect of a different vaccine administration regimen on the immune response. The Protective Immunity from T cells to COVID-19 in Health workers (PITCH) consortium conducted a study on 503 healthcare workers from five U.K. National Health Service (NHS) Hospital centers. They found that SARS-CoV-2 neutralizing antibody titers were higher after the extended dosing interval (6–14 weeks) compared to the conventional three to four-week regimens [22] for the second vaccine dose of an mRNA-based vaccine (BNT162b2 mRNA-Pfizer/BioNTech). The author suggested extending the dosing interval as an effective immunogenic protocol with the added benefit of providing accelerated population coverage with a single dose, as vaccine availability remains constrained [22].

## 3. The Durability of Immune Response

Antibody levels or titers are used as surrogate biomarkers for the efficacy of the vaccine. A drop in antibody levels slowly over time after vaccination is expected. Similar findings have been observed after the administration of COVID-19 vaccines [10]. However, there is insufficient data to suggest that this drop correlates with a decline in protection against the COVID-19 virus (correlate of protection). As a result, the correlate of protection and a protective threshold of the antibody levels for the COVID-19 vaccine is currently unknown.

A study by Widge et al. on mRNA vaccines compared immunogenicity data 119 days after the first vaccination (90 days after the second vaccination) in 34 healthy adult participants. They found a decline in neutralizing antibodies; however, the levels remained significantly elevated (above the baseline) in all participants 3 months after the second vaccine dose. They also found that median neutralizing antibody levels were higher in COVID-19 vaccinated individuals (119 days after the first vaccination) than they were individuals convalescing from COVID-19 with a median of 34 days since diagnosis (range, 23 to 54) [23,24].

Mizrahi et al. analyzed data released by the Ministry of Health in Israel, Maccabi Healthcare Services (MHS). They found that protection against infection and disease dropped over time (from December 2020 to July 2021). They found that the risk for infection was significantly higher for early vaccines compared to those who were vaccinated later. For instance, people vaccinated in January and February were 53% more likely to test positive for SARS-CoV-2 than people vaccinated in March and April [25].

Based on data released by Pfizer–BioNTech, the vaccine's efficacy against symptomatic disease dropped from 96% four weeks after the second dose (first booster) to 84% six months after vaccination [11]. Data from Moderna also showed a decline in efficacy from 94% to 90% after six months of vaccination [9]. It is also crucial to note that the primary goal of the COVID-19 vaccine was to prevent severe disease, and the efficacy of Pfizer–BioNTech and Moderna vaccines against severe COVID-19 remained over 90% after six months [8].

Various other factors may have attributed to this decline in vaccine efficacy other than a fall in immune response, including higher exposure in early vaccinated individuals that tend to be higher-risk population (healthcare workers) or higher testing rate, or mass-media misinformation such as the encouragement of high-risk behavior (not wearing masks in crowded places) in vaccinated individuals.

Due to the lack of reliable protection correlate and threshold, it is difficult to draw conclusions on the need for a third vaccine dose in the general population. In addition, a relatively small decrease in vaccine efficacy should not deter healthcare administrators from prioritizing the primary goal of vaccinating an at-risk global population. A slight decline in titers of binding and neutralizing antibodies over time is expected after vaccination. COVID-19 vaccines may have the potential to provide sustainable humoral immunity. However, further research is needed to determine the protection correlate and protection threshold to address the need for a third vaccine dose.

## 4. Ongoing Trials-Safety and Efficacy

Third doses of the vaccines developed by Moderna, Pfizer–BioNTech, Oxford–AstraZeneca, and Sinovac administered more than six months after vaccination can potentially boost the neutralizing antibody titers, including targets against the Delta variant. In addition, information from these trials suggests that vaccine-related side effects were similar to those observed after the first and second doses of vaccines [26–29]. The safety profile and potential additional protection of a third dose must be weighed against the global scarcity of the vaccine to identify the most vulnerable populations who may benefit from a third dose without jeopardizing the global COVID-19 vaccination effort.

## 5. Vaccine Booster Doses in SARS-CoV-2 Variants

Multiple SARS-CoV-2 variants have been reported globally. Some variants with mutations in the surface spike protein may cause more severe disease, be more transmissible, and

evade vaccine or natural (post-SARS-COA infection-based) immune response. Variants of concern, such as the B.1.1.7 (Alpha), B.1.351 (Beta), P.1 (Gamma, formerly named B.1.1.28.1), and B.1.617.2 (Delta) variants, are more transmissible and cause more severe disease than the wild-type SARS-CoV-2 virus [7]. Fortunately, current COVID-19 vaccines appear to work against the reported SARS-CoV-2 variants. In a study on neutralizing antibodies from vaccinated individuals (vaccine mRNA-127), Pegu et al. found that all individuals had binding antibodies to SARS-CoV-2 variants. Most individuals maintained functional activity against viral variants six months after the second dose of complete vaccination [30].

## 6. Who Could Benefit from the Booster Vaccine Dose

There is limited evidence on the population that may benefit from third or subsequent doses of the COVID-19 vaccine. Of particular interest are immunocompromised individuals, as they may not produce adequate immunological response after two doses of the vaccine. In a study by Kamar et al. on solid-organ transplant recipient immunocompromised patients, they found anti–SARS-CoV-2 antibodies in only 40% (40 of 99 patients) of patients four weeks after dose two. However, that number rose to 68% (67 of 99 patients) 4 weeks after the third dose [31].

On 12 August 2021, the U.S. Food and Drug Administration amended the emergency use authorizations (EUAs) for both the Pfizer-BioNTech COVID-19 vaccine and the Moderna COVID-19 vaccine to allow them to be used for an additional dose in specific immunocompromised individuals who have undergone solid organ transplantation or who are diagnosed with conditions that are considered to have an equivalent level of immunocompromise [32]. According to the CDC, the third dose should be given three weeks after the second dose for the Pfizer vaccine and after four weeks for the Moderna vaccine. As of 7 October 2021, COVID-19 vaccine booster shots are recommended by the CDC for adults aged 65 and up, those who live in long-term care facilities, have underlying medical issues, or work or live in high-risk environments. For example, individuals at an increased risk for COVID-19 exposure and transmission because of occupational or institutional settings such as first responders, education staff, food and agriculture workers, corrections workers, and public transit workers are eligible for a third dose of the COVID-19 vaccine [12].

## 7. Vaccine Booster Doses and Global COVID-19 Vaccine Equity

As many countries gear up to start providing a third (second booster) shot of the COVID-19 vaccine, many experts are questioning the global implications of worsening vaccine inequity [33,34]. The WHO's director-general called it entirely unacceptable for countries that have already used most of the worldwide supply of vaccines to use even more of it [34]. Experts have also highlighted that by ignoring low- and lower-middle-income countries with higher population density and people living in closer approximation, the global efforts to control the COVID-19 pandemic may be futile [35].

In conclusion, third or subsequent doses of COVID-19 vaccines can potentially boost the neutralizing antibody titers against SARS-CoV-2 and its variants, especially in immunocompromised individuals or individuals with underlying comorbidities or who are at an increased risk for COVID-19 exposure and transmission. However, it is vital to exercise appropriate use criteria for additional third or subsequent doses of COVID-19 vaccines without jeopardizing global vaccination efforts and further compounding global vaccine inequity.

**Author Contributions:** Conceptualization, S.P., A.B.S., S.K., and R.S.; data curation, I.G.; writing—original draft preparation, I.G.; writing—review and editing, S.P., S.K., R.S., and A.B.S.; supervision, R.S., and S.P. All authors have read and agreed to the published version of the manuscript.

**Funding:** This research received no external funding.

**Institutional Review Board Statement:** Not applicable.

**Informed Consent Statement:** Not applicable.

**Data Availability Statement:** Not applicable.

**Conflicts of Interest:** The authors declare no conflict of interest.

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
