# Peer review of "COVID-19 Vaccine Booster: To Boost or Not to Boost"

_2036-7449, doi:10.3390/idr13040084_

Round 1

Reviewer 1 Report

The manuscript (review) entitled “COVID-19 Vaccine Booster: To boost or not to boost” briefly covers the basics of COVID-19 vaccines, the durability of elicited immune response, safety, efficacy, and discusses the implementation of a vaccine booster dose. Though various aspects of COVID-19 vaccines have already been published in past years, the information provided in this mini-review is a good resource for scientists involved in vaccine research and for the general audience interested in the current status of the clinical aspects of COVID-19 vaccines. Authors have interestingly discussed the careful consideration for the use of vaccine booster doses. Overall, this manuscript is well written, includes sensible information, and contributes to the existing knowledge.

I recommend authors to work on the following minor comments to improve the quality of the manuscript.

Comment 1: Please cite the relevant reference (line 142-144) for the first sentence under the section “Ongoing Trials-Safety and efficacy”.

Comment 2: Please change the “Covid-19” to “COVID-19” throughout the manuscript.

Author Response

Dear reviewer. Thanks for your suggestions. Please see the response below 

Comment 1: Please cite the relevant reference (line 142-144) for the first sentence under the section “Ongoing Trials-Safety and efficacy”.

Response- Thank you for your kind assessment. We have added the references.

Comment 2: Please change the “Covid-19” to “COVID-19” throughout the manuscript.

Response- This has been corrected.

Reviewer 2 Report

The manuscript by Shekhar et al. attempted to review the “COVID-19 Vaccine Booster: To boost or not to boost. In this review article, the authors demonstrated the importance/drawbacks of the third dose of the COVID-19 vaccine. In brief, the study is new, interesting, compact, and is nicely designed; therefore, the manuscript can be accepted as in the present form.

Author Response

Dear reviewer. Thanks for your encouraging words. In my understanding seems like you have not asked us to do any modification but we have added new data in the paper. 

Thanks again for your time 

Reviewer 3 Report

Dear authors,

There are multiple comments I made, however, I had much more but felt this paper is lacking depth.

From line 42: update the numbers from August to September at least.

No proper citation for Oxford-AZ vaccine (for example, the most appropriate would be, https://www.thelancet.com/journals/lancet/article/PIIS0140-6736(20)32661-1/fulltext)

“Some early studies also suggest that the antibody level triggered by COVID-19 vaccines is falling “ - needs more details, trends, info and citations, e.g. - how and why do they fall? Any differences b/w the vaccines?

“The data on the durability of vaccine response and effectiveness of vaccines against new emerging SARS-CoV-2 variants remains limited “ - there are papers out there, you would need to analyse them and work around them. You would need to cite them as well.

“Despite the lack of concrete evidence, some countries, including China, Russia, Germany, and Israel…” - outdated. The UK implements it as well.

“that cannot cause disease but serves as a vector to delivers “ - should be “to delivery”. English needs a big proper check and proofreading.

“copies of the coronavirus proteins (spike protein)” - are there other ones? Should the reader know?

“After the first dose, immune response (T-lymphocytes and antibody level) slowly 80 drops, leaving behind a small pool of memory B and T cells to guard against future attacks 81 from the same pathogen [15]. A second dose causes a second bigger immune response, 82 which slowly drops off in time. However, a larger number of memory B cells pool (with 83 higher affinity for antigen- affinity maturation) are left behind, helping stage a more ex- 84 tensive and faster immune response against the same pathogen in the future [15]. The 85 Centers for Disease Control and Prevention (CDC) recommends the second dose of the 86 COVID-19 vaccine for Pfizer and Moderna vaccines.” - Strong sentences, only 1 reference (pretty questionable - just a review). If you include that - prove that.

Many other claims are made afterward with a source missing. I don’t want to go into every detail but lines 88-98 have to be revised.

Overall, I had no feeling this paper is bringing any novelty even as a review. It’s questionable whether it will be somehow useful and the lack of proper citations and sources makes it even worse. Sorry for not telling you better news, but in its current state I just want you to dive better and deeper in the literature. And stay up to date.

Kind regards

Author Response

Dear reviewer, Thanks for reviewing our work. We have made changes as per your suggestions. Please see the responses below. 

From line 42: update the numbers from August to September at least.

Response- We have updated the numbers.

No proper citation for Oxford-AZ vaccine (for example, the most appropriate would be, https://www.thelancet.com/journals/lancet/article/PIIS0140-6736(20)32661-1/fulltext)

Response- Thank you for your suggestion, we have added the reference.

“Some early studies also suggest that the antibody level triggered by COVID-19 vaccines is falling “ - needs more details, trends, info and citations, e.g. - how and why do they fall? Any differences b/w the vaccines?

Response- Dear reviewer, we have provided some references in the introduction and expanded on this statement further under the section “The durability of immune response”.

“The data on the durability of vaccine response and effectiveness of vaccines against new emerging SARS-CoV-2 variants remains limited “ - there are papers out there, you would need to analyze them and work around them. You would need to cite them as well.

 Response- Thank you, we have updated the statement.

“Despite the lack of concrete evidence, some countries, including China, Russia, Germany, and Israel…” - outdated. The UK implements it as well.

 Response- Thank you, we have updated it with latest CDC recommendations.

“that cannot cause disease but serves as a vector to delivers “ - should be “to delivery”. English needs a big proper check and proofreading.

 Response- Thank you for your suggestion, this has been modified.

“copies of the coronavirus proteins (spike protein)” - are there other ones? Should the reader know?

  • Response- Thank you, we have modified it.
  •  

“After the first dose, immune response (T-lymphocytes and antibody level) slowly 80 drops, leaving behind a small pool of memory B and T cells to guard against future attacks 81 from the same pathogen [15]. A second dose causes a second bigger immune response, 82 which slowly drops off in time. However, a larger number of memory B cells pool (with 83 higher affinity for antigen- affinity maturation) are left behind, helping stage a more ex- 84 tensive and faster immune response against the same pathogen in the future [15]. The 85 Centers for Disease Control and Prevention (CDC) recommends the second dose of the 86 COVID-19 vaccine for Pfizer and Moderna vaccines.” - Strong sentences, only 1 reference (pretty questionable - just a review). If you include that - prove that.

Response- We have added references supporting above above-mentioned statements.

Many other claims are made afterward with a source missing. I don’t want to go into every detail but lines 88-98 have to be revised.

Response- Thank you for your suggestion, this has been modified.

Thanks again for your time. 

Rahul Shekhar 

Round 2

Reviewer 3 Report

Dear authors,

Thank you for addressing the issues I have highlighted in the previous review. With the corrections made, I am happy to accept the manuscript.